

# Development of a welfare assessment tool for tourist camp elephants in Asia

Raman Ghimire[1], Janine L. Brown[2,3,4], Chatchote Thitaram[1,2,3], Sharon S. Glaeser[5], Kannika Na-Lampang[1], Pawinee Kulnanan[2] and Pakkanut Bansiddhi[1,2,3]

[1] Faculty of Veterinary Medicine, Chiang Mai University, Chiang Mai, Thailand
[2] Center of Elephant and Wildlife Health, Chiang Mai University Animal Hospital, Chiang Mai, Thailand
[3] Elephant, Wildlife, and Companion Animals Research Group, Chiang Mai University, Chiang Mai, Thailand
[4] Center for Species Survival, Smithsonian National Zoo and Conservation Biology Institute, Front Royal, Virginia, United States
[5] Oregon Zoo, Portland, Oregon, United States

Corresponding author
Pakkanut Bansiddhi,
pakkanut.b@cmu.ac.th

## ABSTRACT

**Background:** Approximately one-third of Asian elephants are managed under human care, participating in educational, cultural, religious, and tourist activities. Management conditions vary considerably among venues, raising questions about whether welfare needs are consistently being met, particularly for Asian tourist camp elephants. To evaluate the well-being of elephants engaged in tourist activities, an evidence-based tool is needed for routine assessments to identify potential welfare risks, aid in the development of better camp standards, and enable caretakers to address specific concerns. While many animal welfare tools exist, none have been designed to consider specific environments and management practices faced by elephants living and working in tourist camps.

**Methods:** Using direct observations and interviews, the Elephant Welfare Assessment Tool (EWAT) was developed for tourist camp elephants using the Five Domains Model as a framework. Measures were selected based on peer-reviewed literature, existing standards and guidelines, and opinions from animal welfare experts working with zoo and tourist camp elephants. The EWAT differs from existing tools by including criteria on work activities and restraint methods (*e.g.*, chaining and ankus use), factors common in Asia but not often encountered by western zoo elephants. Measures were scored using a 0–2 Likert Scale. The tool was tested in Thailand and determined by calculating a content validity index (CVI) and conducting inter-rater and test-retest reliability tests.

**Results:** The initial tool included 18 animal-based and 21 resource-based measures across four domains: Nutrition ($n = 5$), Environment ($n = 14$), Health ($n = 10$), and Behavior and Mental State ($n = 10$). Index scores of content validity (CVI) (Item CVI (0.83), Scale CVI/Average (0.98), and Scale CVI/Universal (0.89)) were high. Measures scoring less than 0.83 were removed: the opportunity to mate, the mahout-elephant relationship, and mahout job satisfaction. The final tool consisted of 42 questions related to 36 measures, including 18 animal-based and 18 resource-based measures within the Nutrition ($n = 5$), Environment ($n = 11$), Health ($n = 10$), and Behavior and Mental State ($n = 10$) domains. Intraclass correlation

coefficients (ICC) for inter-rater reliability (0.78–0.90, $p < 0.05$) and test-retest (0.77–0.91, $p < 0.05$) analyses conducted at two camps showed good agreement.

**Conclusions:** This new assessment tool (EWAT) is a context-specific, holistic method designed to offer a practical means of conducting individual and institutional-level assessments of elephant welfare in tourist camps. It is based on the Five Domains Model using reliable and validated animal- and resource-based measures, data collection through direct observation and interviews, and a numerical scoring system. The tool includes several criteria applicable to tourist rather than zoo venues to make it more relevant to the challenges faced by working elephants in Asia.

## INTRODUCTION

Around 15,000 captive Asian elephants live under human care across 13 range countries, predominantly in India, Thailand, and Myanmar. These elephants are used for various purposes, including tourism, education, religious ceremonies, logging, transportation, and forest patrolling (*Menon & Tiwari, 2019*). Thailand alone is home to around 3,800 captive elephants, with 2,700 employed in 250 tourist venues, engaging in shows, riding, bathing, feeding, and observation activities (*Bansiddhi et al., 2018*). This represents more than twice the number of tourist venues for all other range countries combined (*IUCN/SSC Asian Elephant Specialist Group, 2017*). Increasingly, international organizations and animal activists are raising concerns over the welfare of elephants used in tourism (*Duffy & Moore, 2011*; *PETA, 2017*; *Carr & Broom, 2018*; *Baker & Winkler, 2020*), with research studies finding problems with long chaining hours, inadequate exercise, lack of socialization and foraging opportunities, improper equipment use (*e.g.*, ankus misuse, ill-fitting saddles), and provision of high-calorie supplements that leads to obesity (*Magda et al., 2015*; *Norkaew et al., 2018*; *Bansiddhi et al., 2019a*, *2019b*; *Norkaew et al., 2019a*, *2019b*; *Brown et al., 2020*; *Supanta et al., 2022*). While accredited Western zoos must adhere to strict welfare standards, stronger and more enforceable regulations are needed to support evidence-based management standards for ensuring the ethical treatment of tourist camp elephants across Asia (*Bansiddhi et al., 2020*; *Szydlowski, 2022*). Thus, a holistic, evidence-based welfare assessment tool is needed to identify potential welfare risks, inform management decisions, and document welfare changes over time as management changes are made (*Barber, 2009*; *Kagan, Carter & Allard, 2015*; *Sherwen et al., 2018*; *Von Fersen et al., 2018*; *Jones et al., 2022*).

As reviewed by *Ghimire et al. (2024)*, dozens of assessment tools have been developed to monitor the welfare of animals in zoological collections, varying in practicality, reliability, and effectiveness. The comparison of these tools highlighted how challenging it is to comprehensively score factors associated with animal care and welfare because they are multifaceted and often species-specific (*Mason & Mendi, 1993*; *Fraser et al., 1997*). It was

also clear that having well-validated measures and a clear grading system is key (*Hampton et al., 2023*). The most holistic tools rely on both animal- and resource-based measures to record positive and negative affective states (*Kagan, Carter & Allard, 2015*; *Sherwen et al., 2018*; *Von Fersen et al., 2018*; *Benn, McLelland & Whittaker, 2019*; *Jones et al., 2022*; *Baumgartner et al., 2024*). Animal-based measures such as behavior, health, and mental state provide more direct insight into individual animal welfare and how they experience the environment (*Whitham & Wielebnowski, 2009*, *2013*; *Hemsworth et al., 2015*; *Sherwen et al., 2018*; *Benn, McLelland & Whittaker, 2019*). The use of resource measures can identify conditions necessary to support welfare, such as proper housing, food, and water, and is based on the assumption that if appropriate resources are provided, animal welfare will be good (*Whay, 2007*; *Appleby, Olsson & Galindo, 2018*). Access to adequate nutrition, enriched environments, and opportunities for behavioral expression are key drivers of an animal's mental state (*Mellor & Beausoleil, 2015*). The key is ensuring that conditions exist so that captive animals have "agency," or the potential to make meaningful choices with some control over important outcomes in life (*Kagan & Veasey, 2010*).

The most common welfare assessment tools used today were designed for use with multiple species, but there is a growing interest in developing species-specific tools that integrate a comprehensive set of measures that apply to that species (*Jones et al., 2022*). A few elephant-specific tools have been developed for zoo animals, but comprehensive animal- and resource-based measures are often lacking (*Ghimire et al., 2024*). From welfare surveys of hundreds of elephant venues throughout southeast Asia, World Animal Protection has declared that most elephants are kept under poor welfare conditions (*Schmidt-Burbach, Ronfot & Srisangiam, 2015*; *Schmidt-Burbach, 2017*, *2020*). However, those evaluations were based only on resource-based criteria, with low scores given to the presence of an ankus (also known as a bullhook or guide) or if the elephant participated in tourist activities; there were no animal-based measures to corroborate scores of poor welfare. Recently, an assessment tool relying on animal-based measures was developed for routine use by elephant keepers to examine day and nighttime behaviors as indicators of well-being (*Yon et al., 2019*). Zoo-centric tools can be challenging to apply across tourist camp contexts, however. Daily routines in most camps involve tourist activities not common in zoos, such as shows, riding, walking, bathing, and feeding (*Bansiddhi et al., 2018*). In Asian range countries, one or two mahouts are generally assigned to one elephant and are responsible for its care and handling (*Kontogeorgopoulos, 2009*; *Vanitha, Thiyagesan & Baskaran, 2011*; *Mumby, 2019*), whereas zoo elephants are generally managed in groups and by a team of keepers (*Association of Zoos & Aquariums, 2012*; *European Association of Zoos & Aquariums (EAZA), 2020*). Elephants at most accredited zoos are managed in protected contact where humans and elephants are separated by a barrier (*Laule & Whittaker, 2009*), while most camp elephants are managed in free contact and share the same space with mahouts and tourists (*Bansiddhi et al., 2018*). The contact system affects the types of management needed to ensure human and elephant safety, as few tourist venues have enclosures similar to zoo exhibits. Restraint methods like chaining and the use of an ankus are common in Asia (*Bansiddhi et al., 2019a*), but less so in zoos (*European Association of Zoos & Aquariums (EAZA), 2020*). Many large camps manage

more than 30 elephants (*Bansiddhi et al., 2018*), so recording behavior every 30 min throughout the day and night or conducting individual behavior observations over days (*Yon et al., 2019*) is not practical in the context of these tourist camps. Consideration of these fundamental differences is needed in assessing welfare, given the wide variation in management systems used across venues in Asia.

In Western zoos, tracking welfare often utilizes technologies like closed-circuit TV (CCTV), radio frequency identification (RFID), and global positioning system (GPS) monitoring using sophisticated computerized tracking software (*Whitham & Wielebnowski, 2013*; *Whitham & Miller, 2016*; *French, Mancini & Sharp, 2018*; *Coe & Hoy, 2020*). These methods are impractical for use in elephant camps for various reasons because of limited experience in using these tracking technologies, lack of research staff for data input and analysis, budget constraints, and language barriers, as most programs are only in English. Moreover, CCTV monitoring could raise public privacy concerns in tourist camps. Reliance solely on mahout evaluations also has limitations compared to zoo keepers, who are often more familiar with welfare criteria and methodology (*Whitham & Wielebnowski, 2009*). While mahouts are familiar with elephants, they may not consistently identify or understand stereotypic or other abnormal behaviors (*Fuktong et al., 2021*) and present positive biases due to the fear of losing jobs. Thus, expert knowledge should be integrated into the assessment process.

Our goal was to develop a practical, reliable, valid, and low-cost welfare assessment tool for tourist camp elephants in Asia. Validated measures from peer-reviewed literature and expert consultancy were incorporated into the tool criteria, while content validity and reliability tests were conducted to ensure consistency and accuracy in welfare assessments across multiple raters and venues in Thailand (*Jones et al., 2022*; *Malkani, Paramasivam & Wolfensohn, 2022*). Ultimately, it is hoped that the tool will encourage stakeholders to participate regularly in comprehensive evaluations to identify welfare risks and solutions in an under-regulated industry. It also will be key to engaging policymakers to standardize good welfare practices for captive elephants throughout Asia. Lastly, the tool can be used to track welfare changes over time, enabling institutions to make more informed decisions as to what changes are needed to ensure the welfare of elephants across a diverse array of management practices (*Sherwen et al., 2018*).

## MATERIALS AND METHODS

The study protocol was approved by the Institutional Animal Care and Use Committee, Faculty of Veterinary Medicine, Chiang Mai University, Chiang Mai, Thailand (FVM-ACUC; S7/2566) and The Human Research Ethics Committee, Faculty of Veterinary Medicine, Chiang Mai, Thailand (HS5/2566).

### Welfare assessment framework

The Five Domains Model developed by *Mellor et al. (2020)* was selected as the assessment framework and modified to encompass four domains as described by *Racciatti et al. (2022)*: (1) Nutrition, (2) Environment, (3) Health, and (4) Behavior and Mental State.

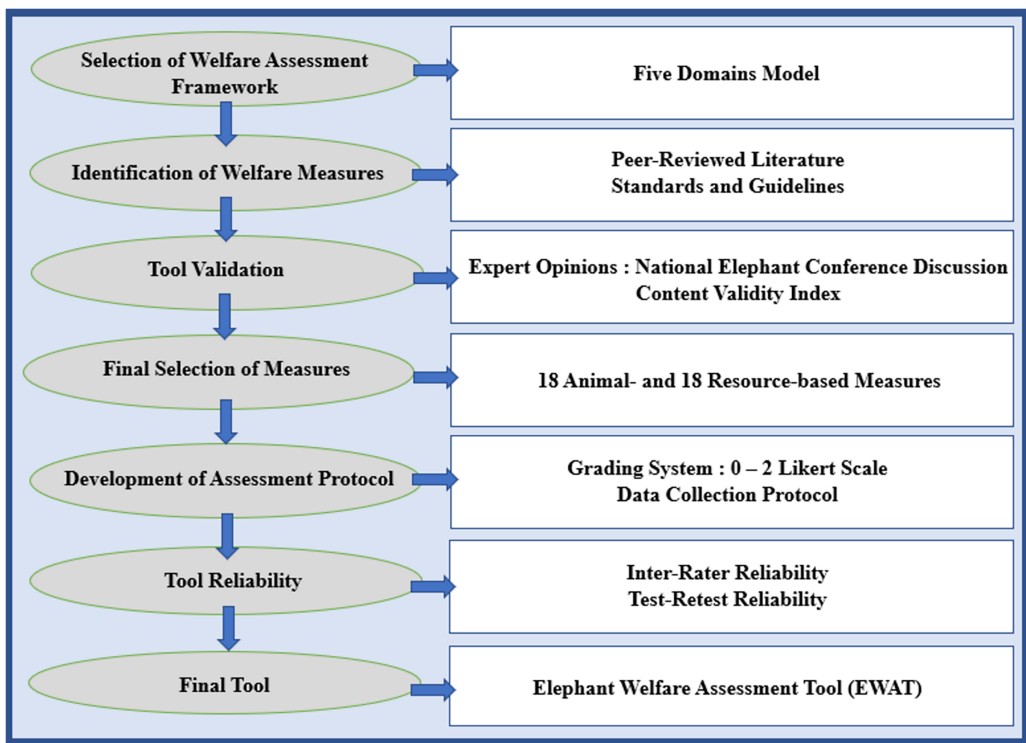

**Figure 1  Steps used to develop the elephant welfare assessment tool for tourist camp elephants.**

The steps used to develop the elephant welfare assessment tool are shown in Fig. 1. First, a review of the scientific literature was conducted to identify relevant elephant welfare measures using PubMed and Google Scholar databases. Search terms included 'elephant', 'Elephantidae', 'Loxodonta', and 'Elephas' used in conjunction with 'behavior', 'nutrition', 'husbandry', 'health', 'welfare', 'living conditions', 'tourist activities', 'housing', and 'mental state'. Guidelines and standards created for elephant tourist camps (Association of British Travel Agents Animal Welfare Guidelines and Usage of Captive Elephants in Malaysia by the Malaysian Association of Zoological Parks and Aquaria) were also reviewed to aid in defining criteria related to resource-based measures primarily.

## Expert opinion input

From the literature review, 39 measures were selected, 18 animal-based and 21 resource-based across four domains: Nutrition ($n = 5$), Environment ($n = 14$), Health ($n = 10$), and Behavioral and Mental State ($n = 10$). These criteria were then reviewed by 46 participants during a Welfare Discussion Session at the 2023 National Elephant Conference held at the National Elephant Institute, Lampang, Thailand. The participants were from government and non-government organizations and comprised 10 veterinarians and staff from the Department of Livestock Development, one staff member from the National Elephant Institute, six veterinarians and staff from the Zoological Park Organization, five veterinarians and staff from the Department of National Parks, Wildlife,

and Plant Conservation, five lecturers and veterinarians from universities, six owners and staff from tourist elephant camps (including one foreigner), and eight veterinarians and staff from animal and elephant foundations. As a result, none of the proposed measures were altered or omitted. However, because tourist camp elephants are managed differently between day and night times, it was decided that six measures (chain length, substrate, hygiene/odor, noise type, enclosure shade/covering, and access to social interaction) would be scored twice, resulting in a total of 45 welfare data points.

## Content validity testing

To further refine the tool and add a statistical dimension to the validation process, we selected six experts (four national and two international, consisting of four elephant veterinarians and two elephant researchers) to evaluate content validity and create a Content Validity Index (CVI). A CVI ensures that the tool accurately measures the intended constructs and encompasses the relevant dimensions of those constructs (*Polit & Beck, 2006*; *Polit, Beck & Owen, 2007*). A step-by-step process was conducted to calculate the CVI (*Yusoff, 2019*) using a form (Article S1) distributed to each expert with instructions to rate the relevance of each measure to the measured domains using a 4-point ordinal scale ranging from 1 (not relevant) to 4 (very relevant). Independent ratings for the 39 welfare measures were obtained and used to calculate both an item-level content validity index (I-CVI) and scale-level content validity index (S-CVI) to assess how accurately the tool reflected the intended construct and assessment purposes (Data S1). Relevance was defined as either 1 (for ratings of 3 or 4) or 0 (for ratings of 1 or 2) for the calculation of I-CVI and S-CVI. The Experts in Agreement metric summed the relevant ratings provided by experts for each item, while the Universal Agreement (UA) metric assigned a score of "1" to items achieving 100% agreement among experts and "0" otherwise. Subsequently, I-CVI was computed for each measure by dividing the experts' agreement by the total number of experts. Measures including the opportunity to mate, duration of the mahout-elephant working relationship, and mahout job satisfaction scored 0.67, 0.67, and 0.50 on a scale of 0 to 1, respectively, suggesting disagreement among the experts as to relevance. Measures including health care, physical fitness/exercise, conspecific interaction, elephant tourist interaction, and stereotypic behavior scored 0.83, while all other measures scored 1, indicating 100% agreement. Measures scoring less than 0.83 were removed (which were three resource-based measures). We also calculated two other scores: the S-CVI/Average, which is the average of the I-CVI, and the S-CVI/UA, which is the average universal agreement score among raters. S-CVI/UA and S-CVI/ Average (Based on I-CVI) were 0.89 and 0.98, respectively, meeting a satisfactory level of content validity (*Polit & Beck, 2006*; *Polit, Beck & Owen, 2007*). CVI was calculated in Microsoft Excel (Microsoft, Redmond, WA, USA).

## Grading system

Welfare scores were based on a Likert 0-2 scale as described in *Baumgartner et al. (2024)* and shown in Table 1. The overall welfare score for each elephant can be calculated by averaging scores across all measures. For camp-level scores, averaging scores across individuals within each camp can be conducted.

**Table 1 Welfare grading system of animal- and resource-based measures.**

| Scores | Animal-based measures | Resource-based measures |
|---|---|---|
| 0 | Welfare concern (Not adequate) | High-risk |
| 1 | Potential welfare concern (Needs improvement) | Moderate-risk |
| 2 | No welfare concern (Adequate) | Low-risk |

**Table 2 Estimated time of each data collection step.**

| Steps involved | Estimated time |
|---|---|
| Interview with the mahout | 15 mins |
| Observation of the daytime rest area | 5 mins |
| Observation of the nighttime rest area | 5 mins |
| Physical examination | 5 mins |
| Behavioral observation (restricted state) | 15 mins |
| Behavioral observation (non-restricted state) | 15 mins |

## Data collection process

A step-by-step data collection process was developed (Table 2) that takes about 1.5 h per elephant. A sampling form (Article S2) was used to record data from interviews with mahouts and camp owners, direct observation of elephants and camp resources, and a physical examination. Elephant behavior was scored twice and randomly in a restrained (chained) and unrestrained state during the day. Care was taken to limit mahout interaction during those observations, and no food was present in the evaluation areas. Each behavioral observation lasted 15 min with at least a 1-h gap in between observations.

## Final tool-elephant welfare assessment tool (EWAT)

The final tool consisted of 36 welfare measures, 18 animal- and 18 resource-based, across the four domains: Nutrition ($n = 5$), Environment ($n = 11$), Health ($n = 10$), and Behavioral and Mental State ($n = 10$) (Table 3). Six measures were scored under two conditions (daytime and nighttime), resulting in a total of 42 welfare data points.

### Nutrition

The Nutrition domain includes measures of feed variety, frequency, freshness, and water frequency and quality. In tourist camps, elephants typically receive roughage as the main dietary source from mahouts on a regular schedule, supplemented with high-calorie foods like bananas and sugarcane from tourists; some are allowed to free-forage for limited periods (*Vanitha, Thiyagesan & Baskaran, 2010*; *Bansiddhi et al., 2018*). As more high-calorie treats are associated with higher body condition scores (BCS), BCS measures in the Health domain can predict the provision of supplements (*Bansiddhi et al., 2019b*). Environment and Behavior and Mental State domains will evaluate access to forage and foraging behavior. Elephants are megaherbivores known for diverse diets in the wild (*Sukumar, 1992*), so a balanced and varied diet is essential for health, longevity,

**Table 3 Types of measures included in the Elephant Welfare Assessment Tool with descriptions of each score, and how data for the measure were collected.**

| Measure | Animal-(ABM) or Resource-based (RBM) Measures | Scoring description | Score | Data collection method |
|---|---|---|---|---|
| **Nutrition** | | | | |
| Feed variety | RBM | 1–2 varieties of roughage/grass | 0 | Interview |
| | | 3–4 varieties of roughage/grass | 1 | |
| | | >4 varieties of roughage/grass | 2 | |
| Feed frequency/Feed availability | RBM | Fixed time, 1–2 times | 0 | Interview |
| | | Fixed time, 3 or more times | 1 | |
| | | Provided throughout the daytime in an unpredictable routine | 2 | |
| Feed freshness | RBM | Pale and dry | 0 | Interview and observation |
| | | Slightly pale and partially green color, slightly moist | 1 | |
| | | Fresh green color and moist roughage | 2 | |
| Water frequency | RBM | Fixed time, 1–2 times | 0 | Interview |
| | | Fixed time, 3 or more times | 1 | |
| | | Water available *ad libitum* | 2 | |
| Water quality | RBM | Turbid, stagnant, or contains dense foreign material (*e.g.,* rotting leaves, plastics, elephant dung) | 0 | Interview and observation |
| | | Clear, but contains foreign material | 1 | |
| | | Clear, no foreign material | 2 | |
| **Environment** | | | | |
| Chain length/Enclosure space (Daytime) | RBM | <6 m chain length or enclosure space of <80 m$^2$ (9 × 9 m) | 0 | Interview and observation |
| | | 6–10 m chain length or enclosure space of 80–315 m$^2$ (9 × 9 to 18 × 18 m) | 1 | |
| | | >10 m chain length or enclosure space of >315 m$^2$ (18 × 18 m) | 2 | |
| Shade (Daytime) | RBM | No shade or covering | 0 | Interview and observation |
| | | Partial covering (*e.g.,* single tree, mesh, or net roof) | 1 | |
| | | Complete covering (*e.g.,* naturally dense canopy, fixed, solid material roof) | 2 | |
| Hygiene (Daytime) | RBM | Dirty area, bad smell, presence of feces within one body length of the elephant | 0 | Interview and observation |
| | | No smell, but feces are stored within 1–5 body lengths of the elephant | 1 | |
| | | No smell, feces are removed regularly and stored >5 body lengths away from the elephant | 2 | |
| Noise type (Daytime) | RBM | Large crowds, direct exposure to traffic or other noise | 0 | Observation |
| | | Occasional crowd noise, little electronic or traffic noise | 1 | |
| | | Only natural sounds | 2 | |
| Substrate (Daytime) | RBM | Concrete | 0 | Interview and observation |
| | | Dirt, grass, or sand | 1 | |
| | | Choice of multiple substrates (Dirt, grass, or sand) | 2 | |
| Access to social interaction (Daytime) | RBM | No direct contact | 0 | Interview and observation |
| | | Physical contact (*e.g.,* trunk, body) with at least one elephant | 1 | |
| | | Can freely interact with one or more elephants | 2 | |

| Table 3 (continued) | | | | |
|---|---|---|---|---|
| **Measure** | **Animal-(ABM) or Resource-based (RBM) Measures** | **Scoring description** | **Score** | **Data collection method** |
| Chain length/Enclosure space (Nighttime) | RBM | <6 m chain length or enclosure space of <80 m² (9 × 9 m) | 0 | Interview and observation |
| | | 6–10 m chain length or enclosure space of 80–315 m² (9 × 9 to 18 × 18 m) | 1 | |
| | | >10 m chain length or enclosure space of >315 m² (18 × 18 m) | 2 | |
| Shade (Nighttime) | RBM | No shade or covering | 0 | Interview and observation |
| | | Partial covering (*e.g.*, single tree, mesh, or net roof) | 1 | |
| | | Complete covering (*e.g.*, naturally dense canopy, fixed, solid material roof) | 2 | |
| Hygiene (Nighttime) | RBM | Dirty area, bad smell, presence of feces within one body length of the elephant | 0 | Interview and observation |
| | | No smell, but feces are stored within 1–5 body lengths of the elephant | 1 | |
| | | No smell, feces are removed regularly and stored >5 body lengths away from the elephant | 2 | |
| Noise type (Nighttime) | RBM | Large crowds, direct exposure to traffic or other noise | 0 | Observation |
| | | Occasional crowd noise, little electronic or traffic noise | 1 | |
| | | Only natural sounds | 2 | |
| Substrate (Nighttime) | RBM | Concrete | 0 | Interview and observation |
| | | Dirt, grass, or sand | 1 | |
| | | Choice of multiple substrates (Dirt, grass, or sand) | 2 | |
| Access to social interaction (Nighttime) | RBM | No direct contact | 0 | Interview and observation |
| | | Physical contact (*e.g.*, trunk, body) with at least one elephant | 1 | |
| | | Can freely interact with one or more elephants | 2 | |
| Environment complexity/ Enrichment | RBM | No enrichment provided | 0 | Interview and observation |
| | | At least one enrichment item (*e.g.*, tree, pole, mud pool, water source) in the environment or mahout-provided (objects to interact with) | 1 | |
| | | >1 enrichment, either in the environment or mahout-provided | 2 | |
| Access to bathing | RBM | No daily baths/showers | 0 | Interview |
| | | Daily bathing by the mahout from a pipe or hose | 1 | |
| | | Daily bathing in the river by free access or by the mahout | 2 | |
| Restriction time | RBM | >12 h/day | 0 | Interview |
| | | 6–12 h/day | 1 | |
| | | <6 h/day | 2 | |
| Access to foraging | RBM | No daily access to foraging | 0 | Interview |
| | | At least once a day | 1 | |
| | | Free access or multiple times a day | 2 | |
| Use of ankus | RBM | Regularly, to establish dominance and cause fear/distress (unjustified punishment) | 0 | Interview |
| | | Regularly, to direct elephant actions or if a verbal command is not effective | 1 | |
| | | Carried but only used in emergencies if elephant actions threaten mahout or tourist safety | 2 | |

(Continued)

| Measure | Animal-(ABM) or Resource-based (RBM) Measures | Scoring description | Score | Data collection method |
|---|---|---|---|---|
| **Health** | | | | |
| Body condition score (BCS) 1 = thin; 5 = fat | ABM | BCS = 1, 5 | 0 | Physical examination |
| | | BCS = 2, 4 | 1 | |
| | | BCS = 3 | 2 | |
| Nail score | ABM | Complicated cracks (nail cracks exposing underlying tissue), overgrowth of nails or cuticles, dry cuticles, infection, moderate or severe injuries, nail loss | 0 | Physical examination |
| | | Uncomplicated cracks (small cracks that do not extend into the cuticle), mild overgrowth of nails or cuticles, mild dry cuticles, mild disfigured nails, or mild injuries | 1 | |
| | | No lesions, normal nails | 2 | |
| Wound score | ABM | Major wounds such as bleeding, infection with pus, deep destruction of tissue, exposed muscle or bone | 0 | Interview and Physical examination |
| | | Minor wounds such as scrapes or scratches, no discharge | 1 | |
| | | No lesions | 2 | |
| Eye condition | ABM | Severe conditions including discharge, ulcers, cataracts, opaqueness, swelling | 0 | Physical examination |
| | | Mild tearing or redness | 1 | |
| | | Clear and bright eye, no discharge | 2 | |
| Skin condition | ABM | Fungal infections, hyperkeratosis, rash, warts/ectoparasites | 0 | Interview and Physical examination |
| | | Cracked or peeling skin, mild hyperkeratosis | 1 | |
| | | Firm and wrinkled skin | 2 | |
| Health care | RBM | No veterinary staff present or available locally | 0 | Interview |
| | | A veterinary assistant/nurse is present, or a veterinarian is on call if needed | 1 | |
| | | Trained veterinarian onsite | 2 | |
| Exercise hours | RBM | <1 h a day | 0 | Interview |
| | | 1–2 h a day | 1 | |
| | | >2 h/day or free choice of movement | 2 | |
| Locomotion/Walking pattern | ABM | Reluctant to move, exhibits evidence of severe pain while walking | 0 | Interview and Physical examination |
| | | Mild lameness | 1 | |
| | | Normal gait | 2 | |
| Urine and feces quality | ABM | Bloody feces, diarrhea, constipation, parasites; bloody or turbid urine | 0 | Interview and Physical examination |
| | | Coarse, dry feces; slightly dark yellow urine, no blood or pus | 1 | |
| | | Normal-shaped, moist fecal bolus; colorless to straw color urine, no blood or pus | 2 | |
| Mucous membrane condition | ABM | Pale or white, dry mucous membrane at the tip of the trunk | 0 | Physical examination |
| | | Pale pink, slightly dry mucous membrane at the tip of the trunk | 1 | |
| | | Bright pink, moist mucous membrane at the tip of the trunk | 2 | |
| **Behavior and mental state** | | | | |
| Foraging/Feeding behavior | ABM | Shows little interest in food provided or foraging opportunities | 0 | Interview and observation |
| | | Consumes food provided by mahout | 1 | |
| | | Consumes food provided by mahout and forages independently | 2 | |

| Measure | Animal-(ABM) or Resource-based (RBM) Measures | Scoring description | Score | Data collection method |
|---|---|---|---|---|
| Rest/Sleep behavior | ABM | Never lies down to sleep | 0 | Interview |
| | | Sometimes lies down, but not every day | 1 | |
| | | Lies down to sleep every day | 2 | |
| Conspecific interaction | ABM | Aggressive to or fearful of other elephants | 0 | Interview and observation |
| | | Shows less interest or avoids interacting with other elephants | 1 | |
| | | Interested, playful, and relaxed with other elephants | 2 | |
| Mahout-elephant interaction | ABM | Aggressive to or fearful of its mahout | 0 | Interview and observation |
| | | Shows less interest and avoids interacting with its mahout | 1 | |
| | | Playful, relaxed, interested in being around its mahout | 2 | |
| Tourist-elephant interaction | ABM | Aggressive to or fearful of tourists | 0 | Interview and observation |
| | | Shows less interest and avoids interacting with tourists | 1 | |
| | | Willing to interact with tourists or no direct tourist interaction is provided | 2 | |
| Elephant general state (Restricted State) | ABM | Chain pulling, violent towards people or other elephants that include kicking, hitting with the trunk, head pushing, charges, head shakes, distress vocalizations | 0 | Interview and observation |
| | | Tense body, head, ear, tail, or trunk; uninterested in surroundings and external stimuli | 1 | |
| | | Relaxed body, head, ear, and tail; regular use of trunk to investigate surroundings and respond positively to external stimuli | 2 | |
| Stereotypies (Restricted State) | ABM | Multiple times a day | 0 | Interview and observation |
| | | At least once a day | 1 | |
| | | None | 2 | |
| Elephant general state (Unrestricted State) | ABM | Tensed body, head, ear, tail, or trunk | 0 | Interview and observation |
| | | Relaxed, alert, responsive, movement of trunk, ear, and tail | 1 | |
| | | Curious and investigates the environment using trunk | 2 | |
| Comfort or Self-maintenance behavior– (Unrestricted State) <br> • Rubbing with a tool <br> • Scratching the body on surfaces <br> • Throwing straw in the body <br> • Body slap with trunk | ABM | None | 0 | Interview and observation |
| | | Shows at least 1 behavior | 1 | |
| | | Shows >1 behavior | 2 | |
| Comfort or Self-maintenance behavior (Unrestricted State) <br> • Water bath <br> • Dust bath <br> • Rolling in mud | ABM | None | 0 | Interview and observation |
| | | Shows at least 1 behavior | 1 | |
| | | Shows >1 behavior | 2 | |

 

reproductive success, and overall welfare in captivity (*Hatt & Clauss, 2006*). Because wild elephants spend 14 to 18 h a day foraging (*Sukumar, 2003*), animals should be allowed to feed during idle times rather than being chained without any feeding opportunities. Increased feeding frequency and unpredictable feeding schedules can also improve body condition, increase activity levels, and decrease food-associated stereotypic behaviors (*Friend & Parker, 1999*; *Rees, 2009*; *Greco et al., 2016*; *Morfeld et al., 2016*) so that was included in the tool. Food quality and nutrient deficiencies often drive elephant behavior and movement in the wild (*Sach et al., 2019*); however, elephants have evolved to survive on a low-quality roughage diet that should be emulated for captive elephants to maintain good gut health (*Hatt & Clauss, 2006*). Such foods also take more time to consume, reducing the behavioral vacuum compared to elephants fed nutrient-dense diets. Feed freshness was included to make sure camps provide fresh roughage as poor-quality diets can cause gastrointestinal problems such as colic and constipation (*Hatt & Clauss, 2006*; *Harris, Sherwin & Harris, 2008*). Similarly, drinking water availability is often limited for tourist camp elephants (*Bansiddhi et al., 2018*), so it was considered an important resource-based criterion in our tool. Elephants have been shown to discern and avoid low-quality water, such as that with high fecal microbial loads (*Ndlovu et al., 2018*). Therefore, ensuring good water quality is essential.

### Environment

The Environment domain includes measures of chain length/enclosure space, shade, hygiene, noise type, substrate, access to social interaction, environment complexity/ enrichment, access to bathing, restriction time, access to foraging, and use of ankus. The opportunity to mate, the duration of the mahout-elephant working relationship, and mahout job satisfaction were included in the initial tool but were removed from the final tool after expert review.

Chaining can cause problems with joints and feet (*Mikota, 2008*; *Buckley, 2008*) and increase stereotypic behaviors (*Gruber et al., 2000*). In the Guidelines on the Usage of Captive Elephants in Malaysia, The Malaysian Association of Zoological Parks and Aquaria (MAZPA), chains should be at least 4 m in length. The Asian Elephant Specialist Group has recommended a chain length of 20 m or an equivalent enclosure space of 1,500 m$^2$ (38 × 38 m). Another report suggests chain length should be at least 20–30 m (*Phuangkum, Lair & Angkawanith, 2005*), which in Thailand is rarely adhered to, with averages of 3 m during the day and 6 m at night (*Bansiddhi et al., 2018*). A few camps do not chain elephants during the day but house them in enclosures at night, although usually in isolation and with limited space (6 × 6 m to 12 × 15 m) (*Bansiddhi et al., 2018*). Because chaining and confinement in an enclosed space both restrict movement, and there are no data on whether one affects elephants more than the other, these criteria were combined. We calculated a minimum enclosure area equivalent to the minimum chain length of 5 m using $\pi \times r^2$ (~80 m$^2$) or about 9 by 9 m. For a score of 2, the length should be more than 10 m and enclosures more than 315 m$^2$ (18 × 18 m). In both scenarios, elephants must have enough room to walk around and, most importantly, lie down to rest.

Access to shade is required for proper care. Asian elephant skin, although thick in some places, is still sensitive to the damaging effects of ultraviolet sunlight (*Schmidt, 1986*), especially on the forehead and ears. Shade is considered adequate if the elephant can freely stand and turn around without being exposed to direct sunlight, and has been associated with the attitude, health condition, and behavior of tourist camp elephants (*Chatkupt, Sollod & Sarobol, 1999*). Adequate sanitation has been defined as the absence of excreta or garbage within a one-body-length radius of the elephant during times of rest (*Chatkupt, Sollod & Sarobol, 1999*), which the expert team did not consider strict enough. Poor hygienic conditions can result in high rates of infection or disease and so should be removed as far away from an elephant as possible. Additionally, elephant feet have soft and sensitive soles (*Gale, 1974*). Chronically wet and dirty conditions can cause foot problems (*Schmidt, 1986*). Proper footing should be on dry, level ground free of debris or dung. Being kept on natural surfaces with a choice of substrates is also important (*Lewis et al., 2010*; *Miller, Hogan & Meehan, 2016*), as foot and joint problems are associated with hard surfaces (*Buckley, 2008*; *Bansiddhi et al., 2019a*). Finally, many camps in Thailand are located just outside of major cities, raising concerns about noise pollution. At least in Africa, elephants have shown risk-avoidance behavior in response to human-generated sounds (*Mortimer et al., 2021*). Elephants have good hearing and use infrasonic sounds for long-distance communication (*Venter & Hanekom, 2010*; *Herbst et al., 2012*), which could be disrupted by human-induced industrial sounds. However, when developing a practical tool for tourist camps, measuring infrasonic interference was not considered. Rather, a subjective estimate of crowd and vehicular noises heard during the assessment was scored.

The importance of providing animals with appropriate stimulation (*Meehan & Mench, 2007*; *Clark, 2017*), choice and control (*Sambrook & Buchanan-Smith, 1997*; *Owen et al., 2005*; *Buchanan-Smith & Badihi, 2012*), and environmental enrichment (*Swaisgood & Shepherdson, 2006*; *Reading, Miller & Shepherdson, 2013*; *Wagman et al., 2018*) has been convincingly documented for a variety of species, including elephants (*Shepherdson, Mellen & Hutchins, 1999*). Enrichment-environmental or provided by care staff  provides stimuli to elicit species-appropriate behaviors that benefit the animal (*Shepherdson, Mellen & Hutchins, 1999*). A complex or enriched environment with a variety of features and objects to interact with is associated with less stereotypic behavior in zoo-housed species (*Swaisgood & Shepherdson, 2006*), including elephants (*Greco et al., 2017*). *Glaeser et al. (2021)* found improved behavioral indicators of well-being in a more enriched and complex zoo environment, with increased activity, locomotion, and foraging levels and decreased stereotypic behavior. Trees and natural water sources allow elephants to exhibit normal behaviors like scratching, dust and mud wallowing, and water play. Providing water for bathing or mud wallowing is also important for skin health (*Domínguez-Oliva et al., 2022*), as well as reducing susceptibility to heat stress due to the lack of sweat glands and a lower volume-to-mass ratio (*Weissenböck et al., 2010*). Bathing also allows for comfort/self-maintenance behaviors and can provide an opportunity for socialization with mahouts and conspecifics. Similarly, access to social interaction is crucial. Social separation has been associated with stereotypic behaviors (*Greco et al., 2017*) and pituitary-ovarian dysfunction (*Brown et al., 2016*) in elephants. Even if animals are unrelated, positive social

interactions can provide a buffering effect against stressful challenges, positively affecting health and well-being (*DeVries, Glasper & Detillion, 2003*; *Plotnik & De Waal, 2014*). While carrying an ankus is often deemed necessary for safety in free-contact situations involving elephants (*Bansiddhi et al., 2019a*), positive training and reinforcement techniques are preferred over punishment-based methods. Elephants are highly sensitive to tactile stimuli, so applying pressure with an ankus elicits avoidance behaviors to move away (*McGreevy & Boakes, 2011*), a common tactic for directing elephant behaviors. Another concern is that some mahouts use nails, knives or other concealed items for the same purpose, which should be included in a welfare assessment. Not all mahouts that carry an ankus use it punitively (*Bansiddhi et al., 2019a*), but the potential for misuse is concerning from a welfare perspective, as incorrect or excessive use could lead to physical (*Bansiddhi et al., 2019a*) and psychological (*Laule & Whittaker, 2009*) harm.

### Health

The health domain includes scores for body condition, nails, wounds, eyes, skin, health care, exercise hours, locomotion/walking pattern, urine and fecal quality, and mucous membrane condition. The BCS index was based on *Morfeld et al. (2016)*, which is routinely used in Thailand (*Norkaew et al., 2018*; *Norkaew et al., 2019a*). Previous studies have shown a strong association between hours of work and BCS, and that increased exercise can reduce the risk of obesity (*Morfeld et al., 2016*; *Norkaew et al., 2018*). Higher BCS has also been linked with altered metabolic status in tourist camp elephants (*Norkaew et al., 2018*). Good foot health, indicated by nail scores (*Ramanathan & Mallapur, 2008*; *Bansiddhi et al., 2019a*) and locomotion/walking patterns (*Harris, Sherwin & Harris, 2008*), is crucial for elephant welfare. However, pad health was not included because few elephants in Asia are trained to lift a foot for examination. As we refine this tool, one objective will be to work with camp owners and elephant veterinarians to emphasize the importance of training this behavior for future health assessments. Wound scoring focuses primarily on human-induced injuries caused by improper use of the ankus or chains, ill-fitting saddle equipment, and resting on concrete floors, which are common welfare issues observed in tourist camp elephants (*Magda et al., 2015*; *Bansiddhi et al., 2019a*). Eye and skin problems are prevalent, necessitating regular monitoring and care (*Ramanathan & Mallapur, 2008*; *Bansiddhi et al., 2020*). Evaluations of other physical factors such as defecation, urination, and eye and skin conditions are best practices that have been noted in guidelines used by the Association of British Travel Agents (ABTA) (*Association of British Travel Agents (ABTA), 2013*). The availability of health care by qualified, trained elephant veterinarians is considered an important part of a welfare assessment (*Association of British Travel Agents (ABTA), 2013*; *Gurusamy, Tribe & Phillips, 2014*). A qualified on-site veterinarian is the foundation of good care and can directly impact elephant health and well-being. Nearly as effective is a consulting veterinarian who makes regular visits to provide preventative care, performs rigorous examinations of all elephants, and is on 24-h call for emergencies.

### Behavior and Mental State

Two domains, Behavior, and Mental State, were collapsed into a single domain similar to *Racciatti et al. (2022)* with measures adopted from *Yon et al. (2019)*. These measures included foraging/feeding behavior, rest/sleep behavior, conspecific interaction, mahout-elephant interaction, tourist-elephant interaction, comfort/self-maintenance behavior, stereotypic behavior, and elephant general state.

Asian elephants in the wild spend most of the day foraging (*Sukumar, 2003*), which often is not supported in captive environments (*Vanitha, Thiyagesan & Baskaran, 2010*; *Bansiddhi et al., 2018*). *Veasey (2020)* defines foraging as the seeking of opportunities to acquire food, which encompasses multiple behaviors and cognitive processes, including walking, sensing and information gathering, decision-making, learning, problem-solving, and socializing. Thus, the ability to do so has implications for the psychological state of elephants. In zoos, studies have shown that providing food in a way that promotes exploratory behavior can reduce stereotypic behaviors (*Swaisgood & Shepherdson, 2006*) and improve body condition (*Morfeld et al., 2016*). Resting behavior is considered a positive welfare indicator (*Chadwick et al., 2017*) with significant impacts on the mental state of elephants (*Williams et al., 2015*). Lower resting rates have been associated with antagonistic social groupings and hard flooring surfaces (*Williams et al., 2015*). Similarly, positive and negative conspecific interaction can reflect on the positive and negative elephant welfare state, respectively (*Yon et al., 2019*).

There is a correlation between mahout welfare and elephant welfare, suggesting that contented mahouts may treat elephants better and with more care (*Vanitha, Thiyagesan & Baskaran, 2011*; *Mumby, 2019*). Studies in zoo settings have shown that positive keeper-elephant relationships are mutually beneficial (*Carlstead, Paris & Brown, 2019*). For example, elephants managed by keepers that rated the relationship positive exhibited lower serum cortisol concentrations, while keepers expressed better job satisfaction. Research has also demonstrated that elephants respond more and faster to behavioral tasks when interacting with mahouts they have known longer (*Crawley et al., 2021*). The interaction of elephants with elephant keepers and conspecifics can reveal both positive (affiliative) and negative (agonistic) welfare states of elephants (*Yon et al., 2019*). Assessing the relationship between elephants and tourists is critical for ensuring both tourist safety and elephant welfare. While tourist camps offer opportunities for intimate and relaxing activities such as feeding and bathing, time allocated to tourist activities can lead to reduced feeding time (*Bansiddhi et al., 2018*). Furthermore, *Ranaweerage, Ranjeewa & Sugimoto (2015)* found that the presence of tourists often triggers increased alertness, fear, stress, and aggression in elephants, and higher tourist numbers have been linked to increased aggression and stress in elephants (*Szott et al., 2019*; *Szott, Pretorius & Koyama, 2019*). Thus, the tool differentiates human-elephant interactions between mahouts and tourists. Stereotypic behavior is commonly used as a welfare measure for captive elephants (*Williams et al., 2018*; *Yon et al., 2019*). However, it is important to note that not all stereotypies describe an individual's current welfare state and that the factors that initially led to abnormal behaviors may no longer be present in the current environment

(*Mason & Latham, 2004*). However, monitoring changes in the frequency or intensity of stereotypic behavior is still valuable (*Mason & Latham, 2004*), especially if coupled with other measures like rest or comfort behaviors, chaining intervals, or social interactions (*Williams et al., 2018*).

The emotional state of an animal, which is indicative of its current welfare experience, is characterized by varying degrees of positive and negative valence (the attractiveness or aversiveness of a situation) and arousal (the individual's level of activation) (*Mendl, Burman & Paul, 2010*). In zoo environments, behaviors reflecting high arousal and positive valence may include courtship displays, allomothering, and social greetings (*Schmid et al., 2001*; *Burks et al., 2004*; *Rose, 2018*), whereas behaviors indicating low arousal and negative valence may manifest as apathy or lethargy due to a lack of stimulation or enrichment (*Mason & Veasey, 2010*). Understanding how individuals respond to the environment is crucial for designing husbandry practices that promote positive valence and arousal levels while minimizing negative experiences. One method of assessing animal behavior to inform on emotional and welfare states is through a Qualitative Behavioural Assessment (QBA) (*Wemelsfelder et al., 2001*; *Minero et al., 2018*). In general, QBA scoring is based on a list of species-relevant terms related to the different dimensions of emotion. QBA measures used by previous elephant welfare assessment tools were incorporated into our tool to assess the elephant's general state in both the restricted and unrestricted states. These measures include: appearing at ease (content), violent (kicking, tusking, whacking with the trunk, head pushing with body, head-on charge) towards others or objects or taking the form of tossing objects about as a displacement activity (frustrated), appearing interested in the environment and/or engaged with, objects or individuals (attentive), head shakes frequent distress rumbles or bellows (distressed), uninterested in the physical environment or social companions (depressed), unrelaxed body, trunk, head postures (distressed) (*Yon et al., 2019*).

## Reliability testing

Reliability testing of the final tool was conducted at two elephant camps outside Chiang Mai, Thailand. Intraclass Correlation Coefficient (ICC) analysis, commonly employed in medical and psychological research (*Weir, 2005*; *McGraw & Wong, 1996*) was used to assess reliability. ICC estimates and 95% confidence intervals (CI) were assessed on an absolute agreement, 2-way random-effects model using the 'irr' package in R Statistics (4.2.2; *R Core Team, 2023*). ICC values of less than 0.5, between 0.5 and 0.75, between 0.75 and 0.9, and greater than 0.90 are indicative of poor, moderate, good, and excellent reliability, respectively (*Koo & Li, 2016*). Results are presented as mean ± SE and range.

### Inter-rater reliability

Inter-rater reliability was calculated for 10 elephants (mean age: 25.9 ± 10.18; male: $n = 1$, female: $n = 9$) rated by three observers who were veterinarians with 3, 6, and 13 years of elephant experience (Data S2). The ICC was 0.82 ± 0.02 (range, 0.78–0.90), indicating good to excellent reliability.

### Test-retest reliability

Test-retest reliability was calculated based on two evaluations of a second set of 10 elephants (mean age: 22.6 ± 13.0 years; male: $n = 3$, female: $n = 7$), 7 days apart by a veterinarian with 5 years of elephant experience (Data S2). The ICC was 0.86 ± 0.08 (range, 0.77–0.91), also indicating good reliability.

## DISCUSSION

This project aimed to develop a welfare assessment tool for elephants managed in non-zoological settings, specifically those at tourist camps in Asia. While many Western zoos are governed by accredited organizations and follow strict standards and guidelines, including requirements for routine welfare assessments (*e.g.*, AZA, EAZA, BIAZA, WAZA, ZAA), such oversight is absent for Asian tourist camps (*Bansiddhi et al., 2020*). Thus, a tool is needed to evaluate the welfare of animals living under conditions that are not generally experienced by those in zoo environments. To that end, the EWAT was designed to be reliable and practical under field conditions and consisted of 18 resource- and 18 animal-based measures to provide information on physical, environmental, behavioral, and social states, as well as husbandry practices used in tourist camps of varying sizes, elephant numbers, and tourist activities.

A range of welfare assessment tools, many developed using the Five Domains Model framework, are utilized by zoological organizations worldwide to conduct regular assessments at individual and/or institutional levels (see review, *Ghimire et al., 2024*). The most holistic tools rely on both animal- and resource-based measures and record positive and negative affective experiences (*Baumgartner et al., 2024*; *Sherwen et al., 2018*). Consequently, the adoption of the Five Domain Model was pivotal to developing our tool specific to elephants in Asia. Like *Racciatti et al. (2022)*, we combined the Behavior and Mental State domains because behavioral indicators directly or indirectly assess some affective states. *Veasey (2020)* has stated that behavior associated with survival, along with species-specific social and cognitive opportunities, are crucial for evaluating the psychological priorities of captive elephants. Thus, this domain was intended to capture behavioral outputs as indices of elephants' perceptions of external circumstances. This integrated approach makes it easier for assessors to focus on and score measures associated with behaviors within a single domain.

The allocation of measures in this new welfare assessment tool addresses a range of biological requirements and is based on recent animal welfare studies (*Bansiddhi et al., 2018*; *Williams et al., 2018*; *Bansiddhi et al., 2019a*, *2019b*; *Norkaew et al., 2019a*; *Yon et al., 2019*; *Brown et al., 2020*; *Veasey, 2020*). However, measures such as safety from predators (*Sherwen et al., 2018*) were omitted because they have a minimal impact on captive elephants in tourist camps. Financial constraints, impracticality in resource-constrained settings like tourist camps, and the aim to create a cost-effective tool also led to the exclusion of physiological measures of glucocorticoid (GC) and immunoglobulin A concentrations (*O'Brien & Cronin, 2023*), which are frequently used to assess the impacts of management factors on health and welfare of zoo and tourist camp elephants (*Brown et al., 2019*; *Edwards et al., 2019*; *Kosaruk et al., 2020*). Likewise, the decision to omit other
invasive methods like blood chemistries or technology-based measurements of heart and respiratory rates (*Wolfensohn et al., 2018*) underscores the commitment to noninvasive and easy-to-apply methodology. However, it is recognized that these techniques can provide valuable insights into the biological functioning of animals, and so should be included whenever possible, if logistically feasible.

In Western zoos, welfare assessments often rely on the use of technologies and keeper knowledge (*Whitham & Wielebnowski, 2009*; *Whitham & Miller, 2016*); however, the practical constraints of tourist camps necessitated a focus more on direct observations and interviews with mahouts. Reliance on keeper ratings to detect subtle behavioral changes can present problems with subjectivity, personal biases, and a lack of knowledge of behavioral diversity (*Sherwen et al., 2018*; *Brouwers & Duchateau, 2021*). Furthermore, a lack of understanding and awareness of animal welfare among mahouts in range countries makes it particularly challenging (*Ward et al., 2020*). To address this issue, our tool integrates the perspectives of the elephant's mahout and an expert rater familiar with welfare metrics and methodology to ensure a more comprehensive and objective evaluation. Additionally, interviews are conducted in the language of the mahout (in this case, Thai), which is vital as most do not speak English. Interviewers can be veterinarians, veterinary assistants, or researchers familiar with elephants. The tool prioritizes ease of use to accommodate the varying schedules and management practices of tourist camps, and to allow mahouts to be actively involved in the assessment process during tourist activities. Similarly, clear communication and coordination with elephant camp managers and raters is essential, especially given the sensitivity of discussing elephant welfare in front of visitors.

Numerical scoring systems are common in welfare assessment tools as they allow for easy data analysis (*Jones et al., 2022*), with scales typically going from 0 to 2 to 0 to 8 (*Kagan, Carter & Allard, 2015*; *Sherwen et al., 2018*; *Von Fersen et al., 2018*; *Benn, McLelland & Whittaker, 2019*; *Baumgartner et al., 2024*). Our tool uses a 0–2 Likert scale, similar to several other welfare tools (*Sherwen et al., 2018*; *Benn, McLelland & Whittaker, 2019*; *Baumgartner et al., 2024*). From a practical perspective, numerical scales are easy to use and allow for rapid assessments. Subjectivity is always a concern, however, especially when using small scales. For example, a 5 m chain length scores a 0, a 6 m chain scores 1; chaining for 12 h scores a 1, while 13 h receives a 0. A revised tool may need to implement a finer scale for measures where little scientific evidence exists regarding how they impact physical or psychological welfare. To that point, selected indicators must be scientifically validated as much as possible or rely on the opinion of field-based experts (*Barber, 2009*; *Melfi, 2009*; *Kagan, Carter & Allard, 2015*). Our expert panel consisted of veterinarians and researchers working with zoo and tourist camp elephants across Western and non–Western countries, who provided input on content clarity, interpretability, appropriateness, and relevance to tourist camps. The calculation of CVI provided a statistical dimension to the experts' opinions, with six experts considered sufficient to calculate a CVI score (*Polit & Beck, 2006*; *Polit, Beck & Owen, 2007*; *Yusoff, 2019*). The resulting high levels of consensus among experts reflected in S-CVI/UA and S-CVI/ Average scores demonstrated the robustness of the tool development and assessment

process. Some measures were revised or omitted after expert review and follow-up team discussions. Notably, the opportunity to mate, mahout job satisfaction, and duration of the mahout-elephant working relationship were removed, while the type of substrate, environmental complexity, use of ankus, and elephant-tourist relationship underwent revisions. The opportunity to mate was removed because experts suggested that females do not always have a choice, with anecdotal reports of forced mating occurring. Although recognized as important to elephant welfare, mahout job satisfaction was removed because experts were concerned mahouts might feel uncomfortable expressing a lack of job satisfaction. Thus, the measure would not be accurate. The high turnover of mahouts throughout Thailand also limits the usefulness of the metric for the duration of the mahout-elephant working relationship. The highest score for 'substrate' was switched from sand to a choice of multiple substrates because of the limited use of sand in relatively small enclosures (*Bansiddhi et al., 2018*) and because giving elephants an option to select a preferred substrate at any given time is ideal (*Meller, Croney & Shepherdson, 2007*). Environment complexity was updated to include mahout-provided enrichment such as food puzzles, balls, and tires. Recognizing the importance of the ankus for the safety of mahouts and tourists when elephants are managed in free contact, a score of 2 was updated to state that the ankus can be carried but only used in emergencies if elephant actions threaten human life. Inclusion of the animal-based measure Wound Score indirectly rates ankus use, as most wounds are caused by misuse of that tool (*Bansiddhi et al., 2019a*). Finally, a score of 2 for the 'elephant-tourist relationship' was revised to reflect scenarios where there is no interaction between elephants and tourists, for example, in camps that only allow observation.

Another aim of this study was to test the reliability of the tool at several tourist camps in Thailand. The ICC values for three raters scoring 10 elephants ranged from 0.78–0.90, which is considered good reliability (*Czycholl et al., 2016*; *Koo & Li, 2016*; *Czycholl, Klingbeil & Krieter, 2019*). This can be attributed to the expertise of the raters, who had 3 to 13 years of elephant experience. Mahouts also contributed invaluable insights based on their working knowledge of elephants. Possible challenges may be encountered in other countries like Nepal, where elephants are managed by two mahouts with distinct responsibilities (*Mumby, 2019*), which could introduce variability in data collection. Elephants were assessed two times for test-retest reliability, which also showed good reliability (0.77–0.91). The assessor, a veterinarian with more than 5 years of experience, performed two assessments within 7 days to limit the influences of seasonality or husbandry changes on elephant behavior and health. It is reasonable to assume that, due to the short timeframe, the assessor could have remembered some scores from the previous assessment; however, remembering all scores of 36 measures for 10 elephants was highly unlikely. While we successfully demonstrated inter-rater reliability and test-retest reliability of the protocol, constraints in staff availability led to test-retest reliability being evaluated by only one assessor. The simplicity in the implementation of the tool likely played a role in achieving good reliability. A straightforward and user-friendly tool enables assessors to apply the measures consistently, regardless of variations in their backgrounds or experience levels. This ease of use minimizes potential errors and misunderstandings

during data collection, which can adversely affect reliability scores. In our study, the simple design of the tool allowed both experienced veterinarians and mahouts to efficiently and accurately assess the elephants, contributing to the high inter-rater and test-retest reliability observed. Additionally, in situations with staff limitations—such as having only one assessor for test-retest reliability—the tool's simplicity ensures that reliable data can still be collected without the need for extensive training or adjustments.

Although the tool was tested by research veterinarians from Chiang Mai University, the tool is simple enough for it to be used by tourist camp staff and mahouts for regular assessments. Plans are underway to involve in-house staff in future reliability testing after they receive training on the welfare assessment process using EWAT. As demonstrated by *Rodríguez Ruiz & Heredia Rico (2013)*, training increases the reliability of results. It reduces protocol application time, so we acknowledge the critical role of proper training in ensuring the quality and accuracy of assessments, as assessor familiarity with elephant behavior is essential, particularly for accurately identifying stereotypic or other abnormal behaviors and conducting physical examinations.

## CONCLUSIONS

Welfare assessment tool development is a continuous process, evolving through refinement and improvement over time as the tool is implemented. In the development of EWAT, the level of detail in the assessment and grading system was defined with consideration for constraints with limited assessment time at each venue and restricted access to observe individual animals involved in tourist activities to avoid interference with venue management. The EWAT represents the first context-specific welfare assessment tool specifically designed for tourist camp elephants. It consists of 36 reliable and valid measures covering aspects of the Five Domains Model with a step-by-step paper-based data collection process. The tool is intended for regular use by elephant veterinarians and mahouts to ensure consistent and ongoing monitoring of welfare as part of in-house assessments. Additionally, it can serve as a useful research tool for elephant biologists and ethologists to track the effects of interventions aimed at improving elephant welfare. A pilot study utilizing the EWAT is currently underway, which should help refine the tool, and provide insights into the tool's ability to assess welfare at individual and camp levels. Welfare score comparisons will also be conducted across domains, parameters such as age, sex, origin, tourist activity, camp size, and daily tourist visits, and between resource- and animal-based measures to facilitate further critical analysis.

Other welfare measures could be used to further validate this tool in subsequent studies. Analyses of GCs are commonly used to validate elephant welfare measures (*Williams et al., 2018*), although it is important to note that GCs are not related only to stress, but are metabolic hormones that respond to stimuli, negative and positive, to maintain homeostasis (*Pokharel & Brown, 2023*). IgA is a novel welfare and health indicator that fluctuates in positive and negative welfare states (*Yeates & Main, 2008*), and so can be equally as difficult to interpret as GCs (*Edwards et al., 2019*). However, both can be measured non-invasively in feces, so incorporating GCs and IgA analyses into the welfare assessment tool could provide additional insight into the biological effects of tourist camp

management factors. However, as summarized by *Pokharel & Brown (2023)*, it is important to consider factors such as age, sex, season, and context for proper interpretation of GC data, all of which can affect and limit the usefulness of this measure, especially for single samples.

Finally, a global effort is required to bring greater awareness and educate people in range countries about animal welfare and the importance of maintaining healthy populations of endangered flagship and umbrella species like elephants. Public perceptions of captive elephants, often shaped by anthropocentric views and a lack of knowledge about animal welfare issues, remain a concern. Institutions and researchers should actively engage with the public, fostering better communication and understanding to help people make informed decisions about which tourist venues to visit. Tourist camps primarily focus on business and economic profits, whereas zoos are more oriented toward animal welfare, species conservation, education, and research. Activities such as shows, riding, logging, participating in religious festivals, and involvement in polo tournaments, as well as management practices about the ankus, chaining, training methods, and weaning provoke international concern. However, the outright dismissal of such practices could lead to tension between local communities and outside experts, and not all activities directly harm elephants (*Brown et al., 2020*; *Kongsawasdi et al., 2021*). Establishing collaborations among all stakeholders is vital for informed tool application and subsequent management adaptations. Such collaborations will also create an environment of open communication about animal welfare among tourist venue staff, benefit staff education, raise awareness, and foster a positive staff culture around animal welfare. The government can play a pivotal role by adopting strong welfare standards, requiring the use of welfare assessment tools and camp audits, and imposing penalties for noncompliance or incentives for compliance. This collaborative approach will ensure that the welfare of elephants is not only assessed but effectively addressed through informed and evidence-based actions.

## ACKNOWLEDGEMENTS

We would like to acknowledge all of the experts including Dr. Alongkorn Mahannop, Dr. Pantep Ratanakorn, Dr. Theerawut Suwathanachao, Dr. Visit Arsaithamkul, Dr. Taweepoke Angkawanish, Dr. Supaphen Sripiboon, Dr. Supissara Wongsuttawas, and Dr. Katie L. Edwards for help developing the tool. We also acknowledge participants of the National Thai Conference 2023 for their suggestions regarding the project. We thank all of the owners, managers, and mahouts at the elephant camps for their cooperation. Special thanks to Dr. Thittaya Janyamathakul, Dr. Panida Muanghong, Dr. Wachiraporn Toonrongchang, and Dr. Pichamon Ueangpaibool for their help and cooperation in data collection.

### Funding

This project is funded by the National Research Council of Thailand (NRCT) to Pakkanut Bansiddhi (Contract number: N42A660900). This study was also supported by Chiang Mai

University, Chiang Mai, Thailand. Raman Ghimire is a graduate student in a Master's Degree Program in Veterinary Science at the Faculty of Veterinary Medicine, Chiang Mai University, under the CMU Presidential Scholarship. The funders had no role in study design, data collection and analysis, decision to publish, or preparation of the manuscript.

## Grant Disclosures

The following grant information was disclosed by the authors:
National Research Council of Thailand (NRCT).
Pakkanut Bansiddhi: N42A660900.
Chiang Mai University, Chiang Mai, Thailand.
CMU Presidential Scholarship.

## Competing Interests

The authors declare that they have no competing interests.

## Author Contributions

- Raman Ghimire conceived and designed the experiments, performed the experiments, analyzed the data, prepared figures and/or tables, authored or reviewed drafts of the article, and approved the final draft.
- Janine L. Brown conceived and designed the experiments, authored or reviewed drafts of the article, and approved the final draft.
- Chatchote Thitaram conceived and designed the experiments, authored or reviewed drafts of the article, and approved the final draft.
- Sharon S. Glaeser analyzed the data, authored or reviewed drafts of the article, and approved the final draft.
- Kannika Na-Lampang analyzed the data, authored or reviewed drafts of the article, and approved the final draft.
- Pawinee Kulnanan performed the experiments, authored or reviewed drafts of the article, and approved the final draft.
- Pakkanut Bansiddhi conceived and designed the experiments, performed the experiments, authored or reviewed drafts of the article, and approved the final draft.

## Human Ethics

The following information was supplied relating to ethical approvals (*i.e.*, approving body and any reference numbers):

The Human Research Ethics Committee, Faculty of Veterinary Medicine, Chiang Mai, Thailand (HS5/2566).

## Animal Ethics

The following information was supplied relating to ethical approvals (*i.e.*, approving body and any reference numbers):

The Institutional Animal Care and Use Committee, Faculty of Veterinary Medicine, Chiang Mai University, Chiang Mai, Thailand (FVM-ACUC; S7/2566).

## Field Study Permissions

The following information was supplied relating to field study approvals (*i.e.*, approving body and any reference numbers):

The Institutional Animal Care and Use Committee, Faculty of Veterinary Medicine, Chiang Mai University, Chiang Mai, Thailand (FVM-ACUC; S7/2566).

## Data Availability

The raw measurements are available in the Supplemental Files.

## Supplemental Information

Supplemental information for this article can be found online at http://dx.doi.org/10.7717/peerj.18370#supplemental-information.

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
