# Peer review of "Development of a welfare assessment tool for tourist camp elephants in Asia"

_PeerJ, doi:10.7717/peerj.18370_

## Round 0.1 · original submission · Major Revisions

Both reviewers regard the study to be excellent and worthy of publication. One reviewer, however, is asking for more tangible results and also raised the issue of repeatability of the study. By addressing these concerns, and those raised by the other reviewer, the paper will be strengthened considerably.

Reviewer 1 ·

Basic reporting

No comment

Experimental design

See general comments below

Validity of the findings

See general comments below

Additional comments

General comments
This work touches on such an important aspect and I really enjoyed working through the manuscript. I do enjoy reviewing papers from which I can learn a lot. The previous reviewers did a compelling job and I couldn’t think of anything to add. The authors did address the reviewer’s comments comprehensively. The paper’s readability is good, and the text has been suitably referenced.
However, I have one critique - it is my understanding that PeerJ is a Science-based Journal, and this paper has very little science in it. It is the longest paper that I have reviewed with so few results. Of course, the nature of the assessment (developing a tool) doesn’t allow for more results. Part of the problem is that there are no controls in place – i.e., if the same exercise was repeated (by the observers and experts alike) by looking at Asian elephants in their natural habitat. Currently, only reference(s) are being made of what one can expect from elephants in the wild. By doing this may provide an opportunity to validate the Welfare Assessment Tool. Of course, not all the aspects of the can be repeated by focusing on wild elephant populations. But it may happen that scoring the different attributes may change (for the captive elephants) if it is evaluated against this suggested form of a control. This doesn’t mean that I doubt the opinions of the experts and the like - by no means – but the suggestion here is to evaluate the actual procedure and by doing that, allow for a comparison (control vs experiment) with statistical tests, and more tangible results.

Reviewer 2 ·

Basic reporting

This manuscript is well-written with relevant literature referenced and background information provided. The EWAT seems to be an important contribution to the field as an assessment tool that is designed specifically for the conditions experienced by many captive elephants in Asia.
I think the order of the steps used to develop the tool could be clarified by making the figure align better with the order presented in the methods section. Matching the terminology used in methods headers would help this as well. The manuscript is a somewhat unconventional format because the methods and results are presented in the same section. For this type of study I think this is ok as long as the section header is updated to reflect this. However if PeerJ prefers them to be separate, the authors need to remove the results from the CVI and reliability analyses from the methods section and put them together in a results section before the discussion.
There are a couple of points that could be made earlier in the manuscript than where they are currently included. First defining/describing what an ankus is the first time it is mentioned would help clarify for readers who may be less familiar with the term. Specifically, you could take the parenthetical statement from 126 and include it instead in line 111. I think that where you mention the ankus in 70 is ok to remain as is though. Second, I think somewhere in the introduction it would be helpful to include that this tool was developed and tested in Thailand since there are several references to Thailand in the methods, but it’s not explicitly stated that the tool was tested in Thailand until the reliability section.

Experimental design

I think that the manuscript presents a relevant and meaningful research question and particularly with the revisions based on past reviewer comments, it is clear how this tool compares to other existing welfare assessment tools. The methodology used to validate the tool is appropriate and supports the authors’ claims for the benefit of this tool for evaluations in elephant camps.

Validity of the findings

The authors have provided all data and the analysis appears statistically sound. Their conclusions are supported by their results.

Additional comments

Some minor line by line revisions:
145 suggest “due to the fear of”
191-193 The 39 measures were already chosen from the literature review, so these were confirmed in the discussion at the conference? I think it would just be helpful to clarify that the 39 measures proposed based on literature weren’t changed from the discussion with stakeholders, since stating this again makes the reader think something is different from the statement in line 182. Maybe the only difference was between day and night measures that resulted from the conference discussion?
200 “international” since national is not plural
220-224 I find the terminology here to be a bit confusing, is S-CVI/Average (based on proportion relevance) the same as S-CVI/UA? You only present two scores in 224 so it seems like these are the same, however I don’t really understand how proportion relevance is part of these scores. From your data provided, one score is the average of the item scores and the other is the average of the UA scores. It would be more clear to simply describe the second score as the average universal agreement instead of talking about proportion relevance.
230 may be better to say “can be calculated”
234 The data collection process here was at first read unclear, I think it would help to clarify that this is the development of the data collection protocol for the tool rather than the data collection process for this entire study. Potentially if you combine the grading system and data collection process into one category/paragraph and use header with something like “Development of assessment materials” this would be more clear. This could also be reflected in Figure 1 where the welfare scoring system and data collection protocol could be the same step.
396-398 This sentence wording is a bit confusing. Is the low resting rate because elephants can’t lie down or have antagonistic social groupings? As it’s worded, it sounds like the low resting rates cause health problems that keep elephants from lying down and prevent the social groupings
492 “are crucial”
511 add a comma after possible
586 I’m not sure how the simplicity in implementation is related to this paragraph, could you expand to explain how it affected reliability?
Table 3- For Stereotypies, I’d suggest changing “No” to “None”
In the future I wonder if the authors would want to consider adding other tools to the questions about ankus use since some mahouts may use different methods of controlling elephants (e.g. nails) when the ankus is not allowed by a camp. I don’t think the authors need to address this comment in the manuscript, but maybe would be relevant for future tweaking of the EWAT.

---

## Round 0.2 · accepted · Accept

The authors addressed all the concerns raised by the reviewer, and the paper can now be accepted for publication.

Reviewer 1 ·

Basic reporting

I am happy with the Authors having addressed all the comments, and I have no further concerns.

Experimental design

No further concerns from my end.

Validity of the findings

Everything here is to my linking, and all the comments have been addressed.

Additional comments

No further comments. Well done to the authors.